# Treatment and Prognosis of Myocardial Infarction Outside Cardiology Departments

**DOI:** 10.3390/jcm10010106

**Published:** 2020-12-30

**Authors:** Anton Gard, Bertil Lindahl, Nermin Hadziosmanovic, Tomasz Baron

**Affiliations:** 1Department of Medical Sciences, Cardiology, Uppsala University, 751 85 Uppsala, Sweden; bertil.lindahl@ucr.uu.se (B.L.); nermin.hadziosmanovic@ucr.uu.se (N.H.); tomasz.baron@ucr.uu.se (T.B.); 2Uppsala Clinical Research Center, Uppsala University, 751 85 Uppsala, Sweden

**Keywords:** myocardial infarction, acute cardiac care, cardiology department

## Abstract

Aim: Our aim was to investigate the characteristics, treatment and prognosis of patients with myocardial infarction (MI) treated outside a cardiology department (CD), compared with MI patients treated at a CD. Methods: A cohort of 1310 patients diagnosed with MI at eight Swedish hospitals in 2011 were included in this observational study. Patients were followed regarding all-cause mortality until 2018. Results: A total of 235 patients, exclusively treated outside CDs, were identified. These patients had more non-cardiac comorbidities, were older (mean age 83.7 vs. 73.1 years) and had less often type 1 MIs (33.2% vs. 74.2%), in comparison with the CD patients. Advanced age and an absence of chest pain were the strongest predictors of non-CD care. Only 3.8% of non-CD patients were investigated with coronary angiography and they were also prescribed secondary preventive pharmacological treatments to a lesser degree, with only 32.3% having statin therapy at discharge. The all-cause mortality was higher in non-CD patients, also after adjustment for baseline parameters, both at 30 days (hazard ratio (HR) 2.28; 95% confidence interval (CI) 1.62–3.22), one year (HR 1.82; 95% CI 1.39–2.36) and five years (HR 1.62; 95% CI 1.32–1.98). Conclusions: MI treatment outside CDs is associated with an adverse short- and long-term prognosis. An improved use of percutaneous coronary intervention (PCI) and secondary preventive pharmacological treatment might improve the long-term prognosis in these patients.

## 1. Introduction

Myocardial infarction (MI) is a condition associated with significant short- and long-term mortality. However, survival after MI has improved over the last decades, both in Sweden and in Europe as a whole, following the development in the management and treatment of the disease [1,2]. Acute and long-term treatment in MI are well documented, with clear, internationally accepted recommendations, including invasive strategies with percutaneous coronary intervention (PCI) and CABG, as well as pharmacological treatment such as anticoagulants, platelet inhibitors, statins, beta-blockers and ACE-inhibitors [3,4]. 

MI treatment also entails certain risks for the patients, including a risk for bleeding complications and kidney dysfunction. These risks increase with higher age and in the presence of comorbidities, wherefore a more conservative approach may be required for some patients [3]. Furthermore, guidelines for MI treatment are developed for treating patients with spontaneous MI that is caused by a thromboembolic event in the coronary arteries (type 1 MI), while evidence-based treatment strategies for MI resulting secondary to other conditions (type 2 MI) are less far developed [5].

MI is associated with serious complications such as heart failure and sometimes lethal arrhythmias. Therefore, a cardiology department (CD) requires a specific structure, equipment and staff training [6]. Still, in some cases, due to, for instance, severe comorbidities or a short expected survival, patients with MI are treated at other departments than CDs. Very few studies have attempted to describe this patient group [7,8]. Therefore, the aim of this study was to describe the MI patients that are treated outside a CD and evaluate if the treatment and prognosis differ in this patient group, compared to MI patients treated at a CD.

## 2. Materials and Methods

### 2.1. Patients

The study was designed as an observational cohort study investigating patents with MI treated outside a CD, in comparison to MI patients treated at a CD. The study comprised consecutive patients discharged with an acute MI diagnose (ICD-code I.21), from eight Swedish hospitals of different size and geographical location, in the year 2011. Since it is mainly CDs that report MI patients to the Swedish national register for MI (Swedish Web-system for Enhancement and Development of Evidence-based care in Heart disease Evaluated According to Recommended Therapies—SWEDEHEART), the study was stratified to also include MI patients outside the registry. The aim was to include the first 100 MI patients reported and the first 100 not reported to SWEDEHEART from each hospital in 2011. This was done by matching the personal ID numbers of all patients discharged with an ICD-code I.21 from each of the hospitals, with the SWEDEHEART registry. However, less than 100 MI patients were found outside the registry at five of the hospitals (Figure 1). The median coverage of Swedish MI patients in SWEDEHEART in 2011 was 88% among patients under 80 years and 60% among older patients [9].

The definition of CD included coronary care units and intermediate cardiac care units. Patients exclusively treated outside a CD are later referred to as non-CD patients, while patients treated at a CD at some point of the care event are referred to as CD patients.

### 2.2. Collection of Data

Data were collected in the same way for all patients, regardless of SWEDEHEART-reporting status. Using a pre-specified case-report form, we collected detailed patient information (including dates of admission and discharge, age, sex, comorbidities, medications at admission, clinical parameters, laboratory results, results from invasive and non-invasive investigations, treatments in-hospital and medications on discharge) from the electronic patient records of each hospital. Copies of ECGs, the discharge summary and other relevant journal notes were also collected.

Survival status and dates of deaths were retrieved from the Swedish Tax Agency.

### 2.3. Troponin Assays

Different cardiac troponin (cTn) assays with different reference values were used at the different hospitals. All original cTn values, reference values and time of sampling were collected; however, all cTn results presented on a group level were standardized by the division of the cTn-level with the 99th percentile upper reference limit of the assay used.

### 2.4. Adjudication of Myocardial Infarction Subtypes

Based on the collected data, the MI diagnosis was retrospectively validated in each case, and all patients were classified as MI type 1–5 or myocardial injury by two independent specially trained physicians, using a pre-specified form based on the Third Universal Definition of MI [10]. In case of adjudication discrepancy, a third physician, and, in a few cases, a fourth physician, was needed to make a majority or a relative-majority decision. Detailed definitions of MI and myocardial injury types are found in Appendix B.

### 2.5. Ethics

This study was approved by the Regional Ethical Review Board Uppsala, reference number 2012/208.

### 2.6. Statistics

Groups were compared by using the Mann–Whitney U test for non-normally distributed continuous variables, presented as medians with 25th and 75th percentiles; Student’s *t*-test for normally distributed continuous variables, presented as means with standard deviation; and Pearson’s chi-square test for categorical variables, presented as numbers with percentages.

Predictors of care outside a CD were analyzed in a multivariate logistic regression model. Variables known at the time of admission that differed significantly between non-CD and CD patients in a univariate analysis were selected for the multivariate analysis (Appendix A). In cases of obvious correlation between two variables, the ones with the most valid cases were selected. Independently significant variables were chosen with backward selection, except gender, which was forced into the final model, presented as a Forest plot. The natural logarithm was used for continuous variables without normal distribution.

Difference in all-cause mortality between non-CD and CD patients was analyzed by using the following univariate and multivariate Cox regression models with stepwise adjustment: 

Model 1—no adjustment. 

Model 2—adjustment for age, sex, active smoking, modified Charlson comorbidity index (1 p for myocardial infarction, congestive heart failure, peripheral vascular disease, dementia, cerebrovascular disease, chronic lung disease and history of bleeding; 1,5 p for diabetes mellitus; 2 p for kidney disease and liver disease; 6 p for metastatic tumor), clinical parameters at admission (heart rate, systolic blood pressure and oxygen saturation), laboratory results at admission (troponin, C-reactive protein, hemoglobin and creatinine) and ECG at admission (STEMI y/n).

Model 3—variables in Model 2 plus in-hospital treatment (intravenous anticoagulants, diuretics, inotropes, antibiotics, invasive or non-invasive ventilation and PCI).

Model 4—variables in Model 2 plus treatment at discharge (renin–angiotensin–aldosterone-system blockers, acetylsalicylic acid, other platelet inhibitors, beta blockers, statins and anticoagulants).

Multiple imputation of missing values for covariates in the Cox regression models was performed (using the SAS function PROC MI and arbitrary missing pattern), with all variables in the covariate section used to produce the values for imputation. Five imputed datasets were used to ensure that the effect estimates were not overly inaccurate due to Monte Carlo variability [11]. The results for each imputation were then combined (using SAS function PROC MIANALYZE). The missing data was considered to be missing at random and the proportion of missing data was highest for current smoking (23.4%; with 41.7% vs. 19.4% missing among non-CD and CD patients respectively) and oxygen saturation (22.2%; with 22.1% vs. 22.2% among non-CD and CD patients respectively). The proportion of missing data in other covariates ranged from zero to five percent.

The survival analyses were performed both with time zero set at the date of department admission and also at 30 days post admission. Follow-up continued until the 15th of March, 2018, when information of deaths was retrieved. 

For all statistical analyses, a *p*-value <0.05 was considered to indicate statistical significance. Logistic regression and crude Cox regression analyses were performed with IBM SPSS Statistics 24.0 (SPSS, IBM Corporation, Armonk, NY, USA). Multiple imputation, multivariate Cox regression and factor importance analyses were performed with SAS Software, version 9.4 (SAS Institute Inc., Cary, NC, USA).

## 3. Results

After examination of the collected data, thirty-nine patients were excluded from further analyses (Figure 1). In 14 cases, the patient had died before arriving at a department. Four of these patients arrived at the hospital with cardiac arrest, and in 13 cases, death occurred at the cath lab. The remaining 1310 patients were included in further analyses.

### 3.1. Departments of Care

A total of 391 patients were initially admitted to other departments than a CD. Out of these, 156 (39.9%) patients were eventually transferred to a CD while the remaining 235 patients were exclusively treated outside a CD. A majority of them were treated at medical departments. Less than one-third were treated at departments with ECG-monitoring and nearly 10% at surgery departments (Figure 2). Information was missing in five patients concerning the department of admission. Among SWEDEHEART reported patients; 10 (1.3%) were treated outside a CD, compared to 225 (42.7%) among the unreported patients. 

In 1075 patients, treatment at a CD occurred at some point of the care event. In 919 of these cases, the patient was initially admitted to a CD; in 1007 cases, the patient was discharged from a CD; and in 10 cases, the patient was treated at a CD in between care at other departments. Among SWEDEHEART reported patients, 773 (98.7%) were treated at a CD, compared to 302 (57.3%) among the unreported patients. 

### 3.2. Clinical Features

The distribution of type 1 MI, type 2 MI and myocardial injury was fairly equal among the non-CD patients. In contrast, a vast majority of the CD patients (74%) had a type 1 MI (Figure 3).

There was a large variety of causes for hospital admission among non-CD patients, with merely 31% presenting with chest pain (Table 1). 

Distinctive clinical features among patients treated outside a CD were high age, female sex and a high prevalence of non-cardiac comorbidities (Table 2). Further, abnormal clinical and laboratory findings were more common among these patients, in comparison to the CD patients. There was no association between the level of the first cTn-measurement and caring department; however, the rise in cTn-levels over serial measurements was more modest among the non-CD patients. Although less common in comparison to the CD patients, ST-elevations on ECG were nonetheless seen in almost 10% of the non-CD patients. Among the non-CD patients; 146 (62.1%) had MI (ICD code I.21) as the main diagnosis, compared with 952 (88.6%) of the CD patients.

### 3.3. Predictors of Treatment Outside a Cardiology Department

Analyzed in a multivariate logistic regression model, older age, metastatic cancer, a depressed level of consciousness, higher CRP and higher creatinine levels on admission were all independently associated with treatment outside a CD. On the contrary, the presence of chest pain and STEMI were both associated with treatment at a CD (Figure 4; the complete model is presented in Appendix A). The most important factors in the model were chest pain and age (Appendix A).

### 3.4. Investigations and Treatment 

The non-CD patients were less often treated with fondaparinux, but more often with low-molecular-weight heparin. Only 3.8% of the non-CD patients were investigated with coronary angiography, and PCI was performed in just four out of the 235 patients. Further, the non-CD patients were prescribed secondary preventive treatment to a lesser extent, with less than one-third of the patients having statin treatment at discharge (Table 3). Among patients surviving the care event, 21% of the non-CD patients were planned for a follow-up to a specialist, compared to 75% among the CD patients. Similar clear differences between non-CD and CD patients regarding treatment and follow-up were seen when comparing exclusively type 1 MI patients in the two groups (Appendix A). In type 2 MI patients, non-CD care was associated with a lower rate of coronary angiography investigations (4% vs. 37.6% in type 2 MI in CDs). Furthermore, non-CD care was also associated with a lesser use of statin and dual anti-platelet therapy for type 2 MI. However, in contrast to the clear differences in type 1 MI patients, the use of renin–angiotensin–aldosterone system (RAAS) blockers, acetylsalicylic acid and beta blockers in type 2 MI did not differ between non-CD and CD patients.

### 3.5. Short and Long Term Survival

The median follow-up was 6.2 years (interquartile range 1.0–7.0 years). During this period, 673 (52.7%) patients died, whereof 203 (87.5%) of those were treated outside a CD and 470 (45%) were treated at a CD (Figure 5). A total of 34 (2.6%) patients, consisting of three non-CD and 31 CD patients, were considered lost to follow-up, since the patients’ personal ID numbers did not find a match in the registries from where survival data were retrieved. This mismatch could be due to foreign citizenship or typing errors. 

The in-hospital, 30-day, 1-year and 5-year mortality among the non-CD patients were 28.9%, 35.2%, 56.9% and 83.2%, respectively, in comparison to 6.6%, 8.4%, 18.4% and 38.4% among the CD patients. The in-hospital mortality also varied depending on SWEDEHEART reporting status. Outside CDs, none of the ten reported patients (0.0%) and 68 of the 225 unreported patients (30.2%) died in-hospital. In CDs, 28 of the 773 reported patients (3.6%) and 43 of the 302 unreported patients (14.2%) died in hospital.

In the total follow-up period, the crude all-cause mortality was significantly higher among non-CD patients in comparison with CD patients, with a hazard ratio (HR) of 3.72 (95% confidence interval (CI) 3.13–4.41). A large proportion of the non-CD deaths occurred early in the follow-up period (Figure 5). The difference in all-cause mortality between the groups was attenuated, but still significant, with a similar pattern in patients > 80 years old (HR 2.05 with 95% CI 1.67–2.51) (Appendix A). As illustrated in Appendix A, the adverse prognosis among non-CD patients, as compared with CD patients, applied to both major MI types; however, it was even more distinctive in type 1 MI (HR 5.04 with 95% CI 3.86–6.57) than in type 2 MI patients (HR 1.85 with 95% CI 1.36–2.51). 

The crude all-cause mortality was higher in non-CD patients both at thirty days, one year and five years after admission (Table 4). The mortality rate remained significantly higher among non-CD patients, also after adjustment for age, sex, active smoking, modified Charlson comorbidity index, clinical parameters on admission, laboratory results on admission and STEMI or non-STEMI with a HR of 2.28 (95% CI, 1.62–3.22) after 30 days, HR = 1.82 (95% CI, 1.39–2.36) after one year and HR = 1.62 (95% CI, 1.32–1.98) after five years. Further adjustment for in-hospital treatment, including PCI, did not eliminate the significant difference in mortality rate between the groups. 

With follow-up starting from thirty days after admission, the adjusted one-year mortality did not significantly differ between non-CD and CD patients. In contrast, the five-year mortality was significantly higher among non-CD patients, also after adjustment for available admission data with a HR of 1.42 (95% CI, 1.08–1.84). However, no significant difference remained after adjustment for either in-hospital treatment (including PCI) or long-term pharmacological treatment.

## 4. Discussion

In this study, MI patients exclusively treated outside a CD were compared with MI patients treated at a CD, regarding clinical characteristics, treatment and prognosis. Treatment outside a CD was associated with advanced age, serious comorbidities, a lesser prevalence of type 1 MI, a lower frequency of PCI and pharmacological treatment to address coronary artery disease, as well as an adverse short- and long-term prognosis.

In total, 235 out of 1310 included patients were treated exclusively outside a CD. However, this study did not aim to study the incidence of MI care outside CDs. In contrast to the inclusion in this study, a vast majority of Swedish MI patients are reported to SWEDEHEART [9], and treatment outside a CD is rare among reported MI patients. Hence, the true proportion of MI patients managed outside CDs is much lower. 

The non-CD patients were most commonly treated at medical departments. In most cases, the MI care was implemented at departments, without the possibility of ECG-monitoring, which implies a deviation from MI monitoring recommendations in the European guidelines [3,4]. One possible explanation, is that only one-third of the non-CD patients had classic type 1 MI, while a majority had type 2 MI or other myocardial injury, for which universally accepted treatment guidelines are lacking [5]. Other possible explanations are advanced age, frailty, and serious comorbidities. Similarly to a Danish study, comparing MI patients treated outside vs. within a coronary care unit, the non-CD patients in the present study were older, more often female and had more non-cardiac comorbidities [8].

A higher frequency of deviating laboratory values and a lower proportion of ICD code 1.21 as the main diagnosis suggest that the non-CD patients, apart from chronic comorbidities, also more often had other acute diseases in need of specialized care elsewhere. A lack of competence in the CDs for handling such non-cardiac acute diseases may explain, in part, why only a minority of the MI patients, initially admitted outside a CD, were transferred to a CD during the care event. Still, 235 non-CD patients were never transferred, despite 62.1% ending up with ICD code 1.21 as the main diagnosis. A short expected survival, contraindications for acute MI treatment or shortage of available CD beds are other possible explanations.

The presence of chest pain was the strongest independent predictor of CD care, according to the multivariate logistic regression and factor importance analysis. Barely one-third of the non-CD patients presented with chest pain, compared with 80% of the CD patients. A relatively high proportion of the non-CD patients had STEMI (10%); nevertheless, STEMI was still independently associated with CD care. A similar, not negligible, prevalence of STEMI was also seen outside the CCU in the Danish study, and like in the present study, the non-CCU patients had lower peak troponin levels but not lower troponin levels upon admission [8]. Hence, the initial troponin level seems to be subordinate to other baseline parameters in deciding where to admit an MI patient. 

The MI investigation and treatment differed clearly between non-CD and CD patients. Important to note, though, is that decisions regarding MI care are routinely made in consultation with cardiologists, regardless of caring department. Investigation with coronary angiography was a rare exception, only performed in 3.8% of the non-CD patients, leading to PCI in just four cases. This may, however, simply reflect that a decision of performing a coronary angiography in an MI patient is a clear indication for admission or transfer to a CD. In comparison, 21 (15.7%) out of 135 MI patients admitted to departments other than a CCU in the Danish study underwent coronary angiography [8]. The higher rate of coronary angiography in that study may follow that only departments of admission were taken into account and not eventual transfers to a CCU, and, further, that the definition of a CD was broader in the present study, also including intermediate cardiac care units.

The non-CD patients were generally prescribed less secondary preventive pharmacological treatment. Thirty-five percent, in comparison with 80% of the CD patients, were prescribed other platelet inhibitors than acetylsalicylic acid, indicating a low level of dual anti-platelet therapy among the non-CD patients. This may partly be explained by a low prevalence of type 1 MI, as well as a higher risk of bleeding due to advanced age, female sex, kidney dysfunction, anemia and, to some extent, a higher prescription of oral anticoagulants [12]. A widespread low-intensity statin therapy could be expected among the non-CD patients due to the higher mean age and higher prevalence of renal impairment. However, it is difficult to justify that merely one-third of the patients were prescribed any statin therapy at all, considering the beneficial effects and safety of the therapy [13]. 

The MIs of the non-CD patients were also followed up, to a low extent, with 21% scheduled for follow-up by a specialist and 28% not scheduled for follow-up at all. This may lead to an underutilization of survival beneficial interventions, such as smoking cessation, exercise-based cardiac rehabilitation, and blood-pressure and drug-adherence control [14,15,16,17]. 

There are MI cases were a conservative treatment approach at other departments than CDs may be appropriate, for instance, in the presence of other acute severe conditions or serious comorbidities. Still there are rarely absolute contraindications for long-term beneficial treatment such as statins, beta blockers or other risk factor interventions. Another possible reason for abstaining from such treatment may be a short expected remaining survival. However, there is likely an undertreatment of non-CD MI patients that needs to be addressed. 

A clear overrepresentation of non-CD patients was observed among patients not reported to SWEDEHEART. The register’s national MI coverage has improved since 2011, from a median of 88% among patients under 80 years to almost 96% in 2019 [9,18]. However, the mean age among non-CD patients was 83 years in the present study, and the coverage of the registry is lower in this age category. This implies that the quality control of the registry does not include non-CD patients to the same extent as CD patients. This may lead to ignorance of deficiencies in MI handling and treatment outside CDs. The quality indices presented in quality registers, such as SWEDEHEART, have the ability to entail great improvements in compliance with guidelines through highlighting differences in patient care [18]. Hence, better routines for quality registry reporting will likely be beneficial for non-CD patients in the long run. 

The non-CD patients had a clearly worse short- and long-term prognosis in comparison to CD patients. This was expected, solely considering the patient selection for CD vs. non-CD care, since an adverse prognosis itself may predispose to non-CD care, as may other serious conditions requiring care at other departments or making acute MI treatment contraindicated. However, a worse short- and long-term prognosis remained in the non-CD group also after adjustment for age, comorbidities and other baseline data, suggesting that factors other than those affecting the suitability for CD care contribute to the adverse prognosis among these patients. Still, no explaining factors could be found for this mortality difference in the analyses, including adjustment for in-hospital treatment such as PCI. However, information on patient characteristics, like frailty or other subtler prognostic factors, as well as department characteristics, including routines, equipment and staff training, was not collected and could not be adjusted for.

In a register-based German study comparing the outcome of MI patients admitted to hospitals with vs. without CDs, a more frequent use of reperfusion therapy and medical therapy (including aspirin, beta blockers and ACE-inhibitors) in hospitals with a CD was associated with a lower in-hospital mortality, despite only minor differences in patient characteristics between the hospitals [7]. The in-hospital mortality was perhaps the most striking difference between non-CD and CD patients in the present study. There are several potential explanations for the high rate of early deaths in the non-CD group. The most important reasons are probably old age and serious comorbidities, whereas a potential direct effect of the MI care outside CDs is probably less significant. However, the results also suggest that patients who died while in hospital were less likely to be SWEDEHEART reported. This may have affected the association between non-CD care and in-hospital mortality, since the patient inclusion was partly conditioned on SWEDEHEART reporting status. Therefore, a greater emphasis should be placed on the survival analyses starting from day 30. The five-year mortality was higher in non-CD patients, as well as among patients surviving the first 30 days, and it remained higher after adjustment for baseline data. However, adjustment for either in-hospital treatment (including PCI) or long-term pharmacological treatment eliminated the significant difference in all-cause mortality between the groups. This suggests that an increased use of recommended therapy outside CDs, which includes use of PCI, platelet inhibitors, statins, beta blockers and RAAS blockers, might improve the long-term outcome of non-CD patients after discharge. There seems to be a clear potential for improving the secondary preventive pharmacological MI treatment outside CDs. However, since patients are probably already carefully selected for PCI, along with the finding that almost none of the non-CD patients received this treatment, it is more difficult to assess both the possibility and potential effect of intensifying the use of PCI outside CDs. 

A limitation of this study is the patient selection, which was partly conditioned on SWEDEHEART reporting status, including 59.8% reported patients, compared to the register’s national MI coverage in 2011 of 88% in patients under 80 years old, and 60% in older patients. This disproportionate share of included patients outside the registry may induce a selection bias (although of less magnitude than a patient selection totally conditioned on SWEDEHEART reporting status). Most importantly, this affects the survival analyses when hospital deaths are included, since both the exposure (department status) and the event (in-hospital survival status) are suspected to affect the probability of SWEDEHEART reporting. However, the survival analyses including only patients surviving the first 30 days should have avoided this bias and be given greater emphasis. Advantages of including patients outside SWEDEHEART are that this patient group covers a larger share of non-CD patients and that unreported patients often are ignored in research, since many Swedish MI studies are based on the registry [2,9,18].

A potential confounding factor when comparing non-CD with CD patients is the different prevalence of MI types in the two groups. Type 2 MI and myocardial injury are both associated with high age, female gender, more cardiovascular and non-cardiovascular comorbidities, a less distinct symptomatology, a lesser frequency of PCI and secondary preventive pharmacological treatment, as well as an adverse outcome, in comparison with type 1 MI [19]. Hence, the lower prevalence of type 1 MI among non-CD patients may explain most of the differences between non-CD and CD patients observed in this study, and it may also be an important reason as to why patients were not admitted to a CD in the first place. However, it would be misleading to exclude type 2 MI and uncertain MI cases when describing non-CD patients, and it is also important to remember that all patients were diagnosed with MI during the care event, including those retrospectively classified as myocardial injury. Sub-analyses on type 1 and 2 MI patients show that the differences between non-CD and CD patients, regarding treatment, as well as survival, were most prominent in type 1 MI patients. Furthermore, there are yet no randomized clinical trials proving a beneficial effect of PCI and secondary preventive pharmacological treatment in patients with type 2 MI and myocardial injury [19]. Therefore, a potential prognostic effect of improving the use of recommended MI treatment in type 1 MI patients outside CDs may rather have been underestimated in the present study, since other MI types were also included in the survival analyses.

Another limitation in the present study is the lack of data on causes of death. Since a large proportion of the non-CD deaths occurred early in the follow-up period, along with the high prevalence of serious non-cardiac comorbidities outside CDs, there is likely a higher proportion of non-cardiac deaths among non-CD patients.

## 5. Conclusions

Advanced age and an absence of chest pain in MI patients predisposed them to care outside CDs. MI treatment in non-CDs was associated with an adverse short- and long-term prognosis. An increased use of PCI and secondary preventive pharmacological treatment might improve the long-term prognosis. Better routines for quality register reporting are needed for the monitoring of the quality of care in these patients.

## Figures and Tables

**Figure 1 jcm-10-00106-f001:**
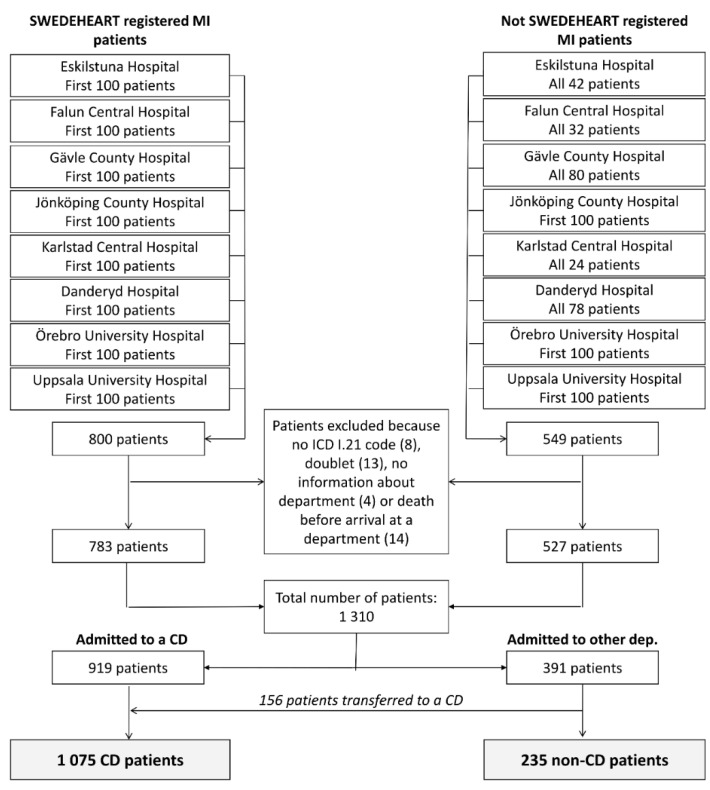
Patient selection process. CD—cardiology department.

**Figure 2 jcm-10-00106-f002:**
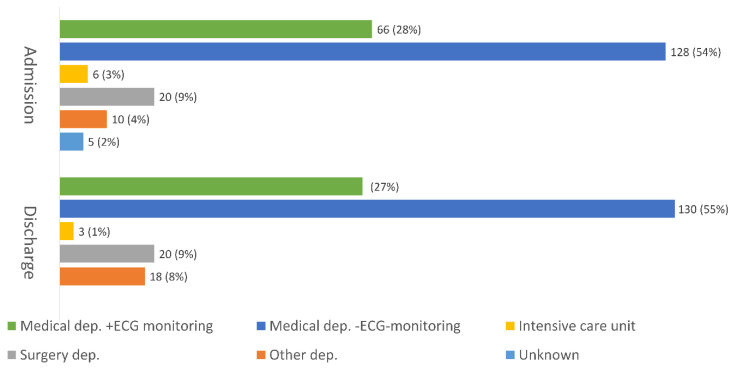
Departments of admission and of discharge among non-CD patients.

**Figure 3 jcm-10-00106-f003:**
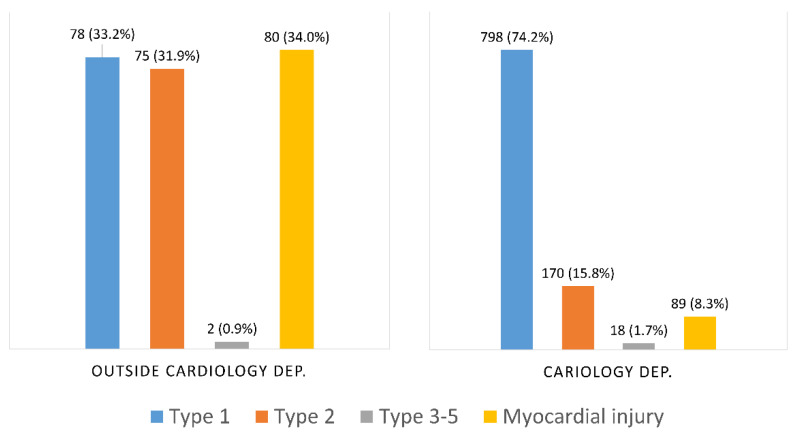
Prevalence of myocardial infarction types outside CDs and at CDs.

**Figure 4 jcm-10-00106-f004:**
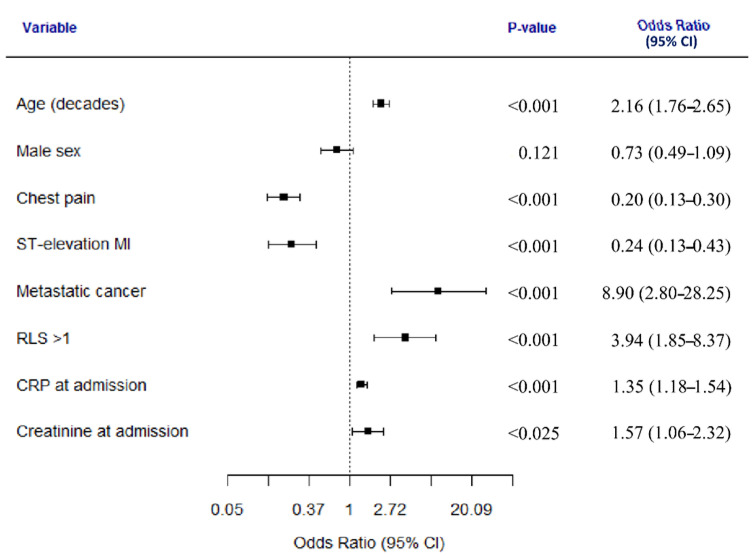
Predictors of treatment outside a cardiology department. Male sex and variables independently associated with treatment outside a cardiology department are presented in the figure. RLS—reaction level scale, CRP—C-reactive protein.

**Figure 5 jcm-10-00106-f005:**
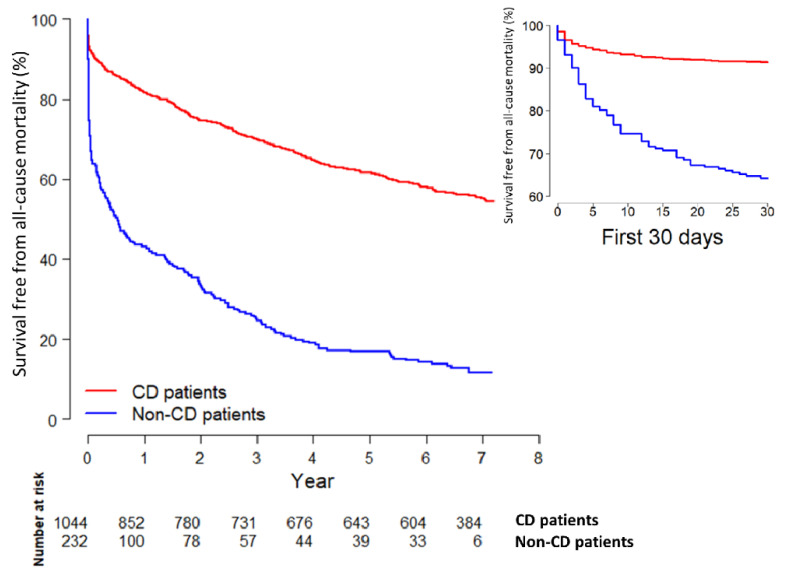
Crude total follow-up and 30 days survival curves for patients treated at a cardiology department (**red**) and patients treated outside a cardiology department (**blue**).

**Table 1 jcm-10-00106-t001:** Causes of hospital admission among non-CD patients.

Symptom (Several Possible)	*n* (%)
Chest pain	73 (31.1)
Dyspnea	68 (28.9)
Fall/fracture	25 (10.6)
Infectious symptoms	21 (8.9)
Reduced general condition	19 (8.1)
Neurologic symptoms	18 (7.6)
Abdominal pain	12 (5.1)
Syncope	11 (4.7)
Pain of other location	8 (3.4)
Confusion	6 (2.6)
Other	23 (9.8)

**Table 2 jcm-10-00106-t002:** Patient characteristics.

	Cardiology Department	Other Department	*p*
Total *n*	1075	235	
Days in hospital, median (IQR)	4 (3–6)	6 (3–13)	<0.001
Age, years, mean (SD)	73.1 (12.5)	83.7 (9.9)	<0.001
Male sex, *n* (%)	657 (61.1)	109 (46.4)	<0.001
Risk factors and medical history, *n* (%)			
Current smoking	191 (17.8)	22 (9.4)	0.002
Diabetes mellitus type 2	215 (20.0)	54 (23.0)	0.306
Hypertension	594 (55.3)	140 (59.6)	0.227
Hyperlipidemia	268 (24.9)	45 (19.1)	0.060
Prior MI	334 (31.1)	86 (36.6)	0.100
Heart failure	162 (15.1)	54 (23.0)	0.003
Chronic kidney disease	79 (7.3)	45 (19.1)	<0.001
History of stroke or TIA	131 (12.2)	51 (21.7)	<0.001
History of major bleeding	16 (1.5)	10 (4.3)	0.006
COPD	75 (7.0)	28 (11.9)	0.011
Dementia	43 (4.0)	39 (16.6)	<0.001
Chest pain at admission, *n* (%)	867 (80.7)	73 (31.1)	<0.001
Clinical findings, median (IQR)			
Oxygen saturation **	97.0 (95.0–98.0)	94.0 (89.3–97.0)	<0.001
Systolic blood pressure, mmHg ***	150 (130–166)	137 (120–167)	<0.001
Heart rate, bpm ***	80 (68–95)	89 (74–100)	<0.001
Temperature > 38°C, *n* (%) *	39 (3.6))	30 (12.8)	<0.001
Reaction level scale > 1, *n* (%) **	28 (3.1)	28 (13.5)	<0.001
Laboratory results, median (IQR) ****			
cTn level at admission, standardized	5.3 (1.8–33.2)	6.1 (1.9–31.1)	0.969
cTn maximum level, standardized	46.2 (12.4–233.1)	23.6 (6.7–122.3)	<0.001
cTn dynamic, %, standardized	392.9 (49.5–2299.9)	140.0 (24.9–893.8)	<0.001
CRP at admission, mg/L	5.0 (2.4–13.0)	26.5 (5.9–77.0)	<0.001
Creatinine at admission, μmol/L	86.0 (71.0–110.0)	101.5 (75.0–150.3)	<0.001
Hemoglobin at admission, g/L	136.0 (123.8–147.3)	124.5 (113.0–133.3)	<0.001
ECG findings, *n* (%)			
ST-elevation	362 (33.7)	23 (9.8)	<0.001

MI—myocardial infarction, TIA—transient ischemic attack, COPD—chronic obstructive pulmonary disease, CRP—C-reactive protein, cTn—cardiac troponin, Tn dynamic—dynamic pattern in troponin levels calculated: ((cTn maximum level–cTn minimum level)/cTn minimum level) x 100. * >65% tested, ** >75% tested, *** > 85% tested, **** >95% tested.

**Table 3 jcm-10-00106-t003:** Investigations, treatment and follow up.

	Cardiology Department	Other Department	*p*
In-hospital, total *n*	1075	235	
Treatment, *n* (%)			
Fondaparinux	540 (50.2)	43 (18.3)	<0.001
Low molecular weight heparin	71 (6.6)	50 (21.3)	<0.001
Antibiotics	159 (14.8)	109 (46.4)	<0.001
Investigations, *n* (%)			
Echocardiography	669 (65.0)	50 (21.3)	<0.001
Coronary angiography	749 (69.7)	9 (3.8)	<0.001
PCI (% of all patients)	593 (55.2)	4 (1.7)	<0.001
Discharge, total *n*	1004	167	
Medications, *n* (%)			
RAAS blockers	765 (76.2)	88 (52.7)	<0.001
Acetylsalicylic acid	930 (92.6)	132 (79.0)	<0.001
Other platelet inhibitors	800 (79.7)	58 (34.7)	<0.001
Beta blockers	876 (87.3)	128 (76.6)	<0.001
Statins	802 (79.9)	54 (32.3)	<0.001
Anticoagulants	90 (9.0)	23 (13.8)	0.046
Scheduled follow-up, *n* (%)			
Specialist	752 (74.9)	35 (21.0)	<0.001
Primary healthcare	183 (18.2)	83 (49.7)	<0.001
No follow-up	46 (4.6)	48 (28.7)	<0.001

PCI—percutaneous coronary intervention, RAAS—renin–angiotensin–aldosterone system.

**Table 4 jcm-10-00106-t004:** All-cause mortality in non-CD patients versus CD patients.

Time Zero	Hospital Admission			30-Days Post Hospital Admission		
Variable	Valid Cases	Hazard Ratio (95% CI)	*p*	Valid Cases	Hazard Ratio (95% CI)	*p*
30-days						
Model 1	1276	4.68 (3.49–6.29)	<0.01			
Model 2	1276	2.28 (1.62–3.22)	<0.01			
Model 3	1276	1.71 (1.19–2.46)	<0.01			
Model 4						
1-year						
Model 1	1276	4.15 (3.32–5.18)	<0.01	1142	3.49 (2.48–4.92)	<0.01
Model 2	1276	1.82 (1.40–2.36)	<0.01	1142	1.38 (0.91–2.08)	0.13
Model 3	1276	1.43 (1.10–1.88)	<0.01	1142	1.13 (0.76–1.70)	0.55
Model 4				1142	1.29 (0.84–1.96)	0.24
5-year						
Model 1	1276	3.84 (3.21–4.58)	<0.01	1142	3.43 (2.75–4.23)	<0.01
Model 2	1276	1.62 (1.32–1.98)	<0.01	1142	1.42 (1.08–1.84)	<0.01
Model 3	1276	1.33 (1.08–1.64)	<0.01	1142	1.19 (0.92–1.55)	0.19
Model 4				1142	1.29 (0.98–1.70)	0.07

Cox regression models for all-cause mortality in all patients (left) and in patients surviving the first 30 days (right). Model 1, unadjusted; Model 2, adjustment for age, sex, active smoking, modified Charlson comorbidity index, clinical parameters at admission, laboratory results at admission and ECG at admission (STEMI y/n); Model 3, as per Model 2, with adjustment for in-hospital treatment including PCI; Model 4, as per Model 2, with adjustment for treatment at discharge.

## Data Availability

The data presented in this study are available on request from the corresponding author. The data are not publicly available due to privacy restrictions.

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
