# Peer review of "Treatment and Prognosis of Myocardial Infarction Outside Cardiology Departments"

_jcm, 2020, doi:10.3390/jcm10010106_

Round 1

Reviewer 1 Report

This well-conceived, -implemented and -analyzed study reports on hospital period, 30-day, 1-yr and 5-yr survival of MI patients (1310 total) that were admitted and treated in facilities that included or not a cardiology department (CD).  Differences in all-cause mortality between CD and non-CD patients were analyzed using 4 regression models with stepwise adjustment for critical variables such as age, sex, co-morbidities, pharmacology and in-hospital treatments.  The authors report that all-cause mortality was higher in non-CD patients including after adjustment for baseline parameters, at 30 days and five years but factors such as age, pharmacology and cardiovascular intervention eliminated statistical significance for the 1-year time point. The authors acknowledge limitations of their study including quality control of the registry used for patient selection, as well as the different prevalence of MI types in the non-CD versus CD groups.  They also relate their findings with other similar studies in Europe.  The authors conclude that MI treatment outside of CDs is associated with adverse short- and long-term prognosis and suggest that improved use of PCI and secondary preventive pharmacological treatment might improve the long-term prognosis of the non-CD patients.  The studies will provide a valuable reference guide, especially for physician in non-CD facilities that are involved in the treatment of MI patients.

The study is clearly described, well-written and the results support the conclusions.  I have the following suggestions for the authors to consider:  If possible, I believe it would be of value if the authors could include cause of death data, even limited to cardiac versus non-cardiac, considering that the patients in the non-CD group die earlier and more frequently, but have more extreme co-morbidities - even better link this with age/survival curves for both groups.   What is the primary reason for the massive early death rate in the non-CD group?  Also, and somewhat related, is it possible to distinguish the effects of PCI versus pharmacology on outcomes?  Finally, whereas the authors show that the differences (30-day, 5-yr) for the most part remain highly significant after correction for age, because the age differences in the two groups is large (>10-years) and most of the non-CD deaths occur early, I wonder what the data would look comparing for example only patients >80 years in both groups?

Reviewer 2 Report

Dear authors,

I read your study on MI treatment in CDs and non CDs with great interest. I shows that even though we might often think that our treatment of "standard" diseases is up-to-date in all areas of our countries, there is still much room for improvement. Especially, this might be the case for more decentralized areas - these are often also those not covered by large studies done in big cities and/or tertiary hospitals.

While the manuscript is very interesting, I still have some concerns:

-) overall manuscript: Please revise the whole manuscript in terms of English style and grammar. For example, there are already a few sentences that sound not right in the abstract (e.g., lines 15-17).

-) Abstract, line 15: I would change to "all cause mortality".

-) Abstract, line 15: I would change to "until 2018" (the exact date seems unnecessary in the abstract).

-) Introduction, line 32: lose "the" before "survival".

-) Introdcution, line 37: change to "entailS".

-) Introduction, line 38: "these risk increase" (no s).

-) Introduction, line 42: I would change to "are less far developed".

-) Introduction, lines 43-45: Rephrase sentence.

-) Introduction, lines 45-46: Give examples of why MI patients might not be treated at CDs in some cases (e.g., isolated areas,...).

-) Methods, patients: How was a CD EXACTLY defined (in detail!)? What about Emergency Departments?

-) Methods, patients: I still do not quite get how you acquired your data. Sweden's CDs report their MIs to a registry, but how exactly did you acquire data from non-CD-MIs? And how did you overcome the bias of non-reporting? (not-so-severe or on the other hand very severe cases might not have been reported, therefore a large number of MIs in both CDs and non-CDs might be missing in your cohorts).

-) Results, line 147: How come so many patients with (apparent?) MI symptoms were treated at departments without ECG monitoring and even at surgical departments? Were they diagnosed there and then refered, or really treated there? I think if they were really treated there, than you compare standard of care with non-standard of care and your results are heavily influenced by that. At least, this must be reflected in your Discussion, Limitations, and Conclusion (and also in your abstract). Also, why have you not adjusted for department type in your regression analyses?

-) Results, line 161: Similar to my previous comment about the departments, I think that if most MI type 1 cases were treated at CD departments, all other MI types might have been diagnosed and treated slower (because not so apparent), and therefore this might have influenced results? Also, why have you not adjusted for MI type in your regression analyses?

-) Line 228 (Figure 5): Please change to "all cause mortality" here, also in the whole text.

-) Line 283-285: Again, I am very surprised about this lack of referal, please go into detail about why this might have been the case.

-) Conclusion, last sentence: Please rephrase, the sentence is quite confusing.

Round 2

Reviewer 2 Report

Dear Authors,

Thank you for addressing all my comments and concerns. I feel that the manuscript now has improved significantly. Please run another English spelling and grammar check.